# Quality of Life in Low-Risk MDS: An Undervalued Endpoint

**DOI:** 10.3390/jcm11195699

**Published:** 2022-09-27

**Authors:** Bert Heyrman

**Affiliations:** Department of Haematology, Ziekenhuis Netwerk Antwerpen (ZNA), 2020 Antwerp, Belgium; bert.heyrman@zna.be

**Keywords:** myelodysplastic syndromes, quality of life

## Abstract

The opinion I put forward in this paper is that attention must be paid to health-related quality of life as a study endpoint in lower-risk MDS patients. At the moment therapeutic options are limited in this population. New treatments are predominantly available in clinical studies. In announcing trial publications and during manuscript introductions, quality of life is widely valued as a treatment goal. However, data on health-related quality of life during phase III studies are not published in the original publications, thereby undermining the importance of quality of life as a study endpoint. What seems to be forgotten is that quality of life comprises more than a study endpoint. It is the highest good of lower-risk MDS patients and should also be acknowledged as a safety aspect. Current phase II trials with new medications do not collect data on health-related quality of life, a practice that I consider unethical. In this opinion I demonstrate the current attitude towards health-related quality of life in lower risk MDS patients among leading journals and trial sponsors with several recent examples. I also argue that health-related quality of life should be the main treatment goal in this population. In the event that we shift our focus towards health-related quality of life as the main treatment goal, new treatments could come to a field where gains in overall survival have been marginal over the years.

## 1. Surrogate Endpoints Divert Our Treatment Focus

Health-related quality of life (HRQoL) is defined as an individual’s perceived physical and mental health over time. This comprises a standalone parameter that concerns every human being and will not easily be abandoned. Only for the cause of survival benefit are we willing to accept a diminished HRQoL. To what extent this diminishing is acceptable depends on the time we need to put up with a diminished HRQoL and the magnitude of the reduction. What is acceptable differs from person to person. In general, we prefer the shortest period possible and smallest reduction in HRQoL and wish for a full recovery afterwards. In incurable diseases, HRQoL becomes a more prominent issue. This is due to the fact that there will be no healthy phase following treatment, and therefore no full recovery of HRQoL is expected. In the case that a new treatment negatively impacts HRQoL, a clear benefit must be obtained in overall survival that makes it worth going through the treatment. If a positive effect on HRQoL is seen, a gain in overall survival is not mandatory. This approach has proven to be successful and has dramatically changed how we treat myelofibrosis [1]. For low-risk myelodysplastic syndrome (MDS) patients, new treatment options focus on lowering transfusion as the primary objective. This results from the impact of transfusion rate and secondary iron overload on progression-free survival [2]. The final impact on overall survival of newly installed treatments has yet to be shown. If lowering the transfusion rate can be achieved, announcing an amelioration of the transfusion burden is kicking in an open door. Whether or not this has a positive effect on health-related quality of life should be the main question.

## 2. HRQoL Should Be the Primary Treatment Goal and Needs a Standardized Unit of Measurement

Lower-risk myelodysplastic syndromes are known to occur predominantly in the elderly population, who are not eligible for allogeneic transplantation. In this setting the condition is uncurable. The median age at diagnosis of myelodysplastic syndromes is 71 years old [3]. The predicted overall survival in the IPSS-R very-low-risk group is 8.8 years [4]. With an overall life expectancy of around 80 years, there is a small margin of profit in terms of overall survival in the low-risk MDS population. Therefore, health-related quality of life (HRQoL) is a primary treatment goal of installed treatments. The acknowledgment of HRQoL as a treatment goal has led to disease-specific HRQoL questionnaires. This is important because a quantitative measurement of HRQoL allows for a standardized follow-up during a treatment course and measurement of the treatment’s impact on HRQoL. The use of standardized questionnaires goes further than serial evaluation of HRQoL; it gives us the possibility to look across different trials. This will influence our treatment choice when more treatment options become available. A consensus on a standardized unit of measurement is needed. The most developed and currently used options are the QOL-E© and QUALMS [5,6]. It is time to commit to these tools and avoid the struggling through four or five questionnaires trial patients are currently subject to.

## 3. Oversharing Is a Pitfall, Overtreatment Should Be Avoided

In clonal haematopoiesis of indeterminate potential (CHIP), per definition the patients do not meet the criteria for MDS, and therefore the disease itself is asymptomatic. There can only be a psychological impact on HRQoL from the moment we communicate this condition to our patients. It remains part of the art of medicine to estimate whether the patient would like to know this or not and therefore difficult to outline in a uniform guideline. Disease evolution in MDS and overall survival largely depend on the mutational profile of the disease [7]. Any treatment that affects this unstable environment demands extra caution because progression to acute myeloid leukemia is always lurking. The perception remains that new installed treatments will only augment or stabilize HRQoL during treatment. Deterioration in quality of life as a result of an installed treatment appears implausible. As we have seen in several treatments in other haematological conditions, the mode of administration can change over time [8]. We strive for subcutaneous injections over intravenous administration, low frequency over high frequency and administration at home over administration in a hospital. If we truly want to understand what these interventions mean for our patients, a uniform collection of HRQoL data is needed. When a patient is classified as having low-risk MDS, any treatment decision should be made carefully, as the patient will have to live with this decision for many years.

## 4. HRQoL Is Undervalued as a Study Endpoint in Publications of Phase III Studies

A prerequisite to make an evaluation of HRQoL during treatment and perform a correct overarching analysis is publishing HRQoL data in the original publications. In the original publication of the Medalist trial, treatment goals in lower-risk MDS patients were clearly stated during the introduction, i.e., transfusion independence, improvement in haemoglobin levels and maintenance of or improvement in quality of life [9]. The manuscript reports widely on the first two goals but fails in reporting quality of life data, although they were available [10]. Due to the HRQoL results not being published, this domain remains open for speculation. This can be misleading due to positive results in terms of erythroid response that bring a positive flare to all the treatment goals. Publication of HRQoL data in this specific trial was diverted to another journal with a lower impact factor, confirming the attitude of the journal towards HRQoL as a study endpoint during this trial.

Another phase III study that investigated the use of oral azacytidine (CC-486) in lower-risk MDS patients acknowledged the importance of quality of life as a treatment goal in the announcing manuscript and collected data accordingly [11]. The trial exposed the occurrence of more early deaths in the CC-486 arm compared to the placebo group, with a similar overall death rate, and concluded further investigation is needed. In terms of HRQoL outcome, the authors inserted a small paragraph announcing positive results. They committed to reporting these results in full detail ‘elsewhere’ [12]. So far, these positive results on HRQoL have not been published. The trial confirms that CC-486 has no positive impact on overall survival, but the journal missed the opportunity to extensively report on what was announced: positive HRQoL data. This undermines HRQoL as a treatment goal and weakens motivation for further analysis of CC-486. In case a clinically significant amelioration in HRQoL could be demonstrated, this treatment is worthy of further investigation and could become a future option even in the absence of overall survival benefit.

## 5. HRQoL Is Neglected as a Safety Aspect during Phase II Studies

Apart from journals, it appears that pharmaceutical compagnies too have not captured the importance of HRQoL in this patient population. Data on HRQoL are not collected during earlier study phases. In the publication of the two-part phase II/III study of imetelstat, another upcoming treatment for low-risk MDS patients, no quality-of-life data were reported [13]. To the best of my knowledge, no HRQoL data were collected during this phase II study of imetelstat. A similar observation was made in the publication of the phase II study of rigosertib [14]. Any treatment in lower-risk MDS with negative impact on HRQoL may not be seen as safe. During phase II studies, safety aspects should be a primary concern apart from treatment efficacy. Any impact on quality of life should at least be monitored as a safety aspect and always be implemented in safety reporting in the original publications. Not collecting data on HRQoL in earlier study phases is therefore unethical.

## 6. Shifting the Focus Outcome to HRQoL Will Open the Field for More Treatment Options

Only a handful of treatments have shown any benefit in lower-risk MDS patients during the last 2 decades. The heterogeneity of the disease is seen as the culprit for the limited success. Perhaps we were focusing in the wrong direction, making treatment success nearly impossible. In the absolute knowledge that life is never endless, we should shift our focus towards HRQoL as the most important treatment goal. This will motivate us to look at new treatments from a different perspective. Treatment success will no longer be measured by surrogate endpoints such as transfusion rate or transfusion burden. We will hereby open the door for new molecules that could previously not be linked to the disease to come to the field of MDS.

We may only be content in the case that we succeed in obtaining a meaningful amelioration in quality of life for our low-risk MDS patients. This opinion is warm call to the MDS community to value the health-related quality of life of our lower-risk patients and not accept the minimization of its importance by leading journals or pharmaceutical companies who primarily use large trials.

## 7. Conclusions

From the moment a patient receives a diagnosis of lower-risk myelodysplastic syndrome, he/she will most likely suffer from a life-long disease that is beyond our impact on overall survival. Quality of life is widely accepted as a primary treatment goal in this population and should at least be valued as an equivalent study endpoint to transfusion independency. Apart from a treatment goal, HRQoL should also be acknowledged as a safety aspect, and therefore data should be collected during earlier study phases. With new molecules coming to the field, it is the responsibility of trial initiators to collect data on HRQoL using standardized methods such as the QOL-E and QUALMS also in earlier study phases, and it is the responsibility of publishing journals to share these data in the original publications, regardless of their outcome.

## Data Availability

Not applicable.

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
