# Peer review of "Quality of Life in Low-Risk MDS: An Undervalued Endpoint"

_jcm, 2022, doi:10.3390/jcm11195699_

Round 1

Reviewer 1 Report

In this opinion letter, Dr. Heyrman has pointed out an interesting and also unoticed aspect of hematological malignancies with long life expectancies, which are not only low risk myelodysplastic syndromes (MDS), but also chronic lymphocytic leukemia or chronic myeloid leukemia, or myelofibrosis. In this view, it will be worthy to mention studies and strategies for improvement in quality of life (QoL) in these diseases too. However, MDS remains a black hole for QoL, especially with these novel clinical entities of uncertain significance; it will be nice to hear your opinion also on this aspect: what is the QoL of a subject who has received a diagnosis of CHIP? What would we should do in terms of QoL for these subjects?

I would suggest to revise the CC-486 clinical trial part as it is very negatively discussed. Also try to highlight the good of that trial.

Another point that I would like you to discuss and have your opinion is related not only to therapeutic strategies, but also on how subcutaneous or i.m. or e.v. administrations can impact QoL of low risk MDS patients. What do you think about the need to regularly take by s.c./i.m/e.v. growth factors even at home and how this could impact QoL of these subjects.

Author Response

A referral was made to the success of ruxolitinib as a treatment option in myelofibrosis. This strengthens the argumentation of HRQoL importance.

CHIP is a difficult entity to score in terms of quality of life. Per definition the patients do not meet criteria for any haematological malignancy and therefore the disease itself is asymptomatic. There can only be a psychological impact on HRQoL from the moment we communicate this condition to our patients. In my opinion it is the ‘art of medicine’ to estimate whether the patient would like to know this or not and therefore difficult to outline a uniform guideline.

The paragraph of CC486 has been rephrased.

In the ideal world we have different modes of administration available and we use them by patient preference. As we see in the administration of immunoglobulins, some prefer a weekly SC injection of several hours at home and others prefer a monthly administration IV in the hospital. I implemented a new paragraph related to caution for overtreatment and referred to different routes of administration.

Reviewer 2 Report

In this opinion article, the author advocates health-related quality of life (HRQol) as a treatment goal in lower-risk MDS. The author briefly reviewed the concept of HRQol and urge physicians to find the balance between HRQol and OS. Furthermore, the author mentioned a few major clinical trials where QoL results were not presented that otherwise could be useful to assess outcome independent improvement of QoL. Then the author discussed the importance of improving QoL in cases where only marginal gain can be achieved in OS. Finally, the author urged physicians and pharmaceutical companies to include QoL as an equivalent endpoint in addition to transfusion improvement. While the message the author conveys is loud and clear, the overall flow of this paper can be more organized and fulfilled as suggested below.

It would be better if the author summarizes a take-home message at the beginning of each paragraph/section.

In the first paragraph line 24-33 and line 76-92 discuss the balance between clinical response and the importance of assessing QoL when clinical response/outcome cannot be significantly improved and therefore can be merged.

Assessing HRQoL is very challenging as there is no consensus on a standardized questionnaire or system internationally and that also varies between trials since the therapeutic agents are different. The author should expand on this part, particularly naming any HRQoL assessment system that could potentially be applied to LR-MDS trials and cite trials that have successfully transformed the practice of physicians by including HRQoL.

HRQoL should be consistently used, line 29, 30, and 32 should be corrected. HRQoL and QoL were used in a mixed way which may confuse the readers.

The author may want to discuss over-treating LR-MDS may accelerate AML transformation. 

Author Response

Each paragraph is now preceded by a take home message.

The balance between clinical response and QoL is now discussed in one paragraph. A very short repetition is kept, since I felt it was important in the paragraph to accentuate HRQoL as a safety aspect.

A recommendation is made to use two standardized questionnaires and commit to a standardized unit of measurement. 

A referral was made to the success of ruxolitinib as a treatment option in myelofibrosis based in HRQoL improvement. 

HRQoL is now used consistently through the manuscript.

A new paragraph was implemented with a warning for overtreatment, referring to possible evolution to AML.